# PathAlign: A vision–language model for whole slide images in histopathology

**Faruk Ahmed**[1] **Andrew Sellergen**[1] **Lin Yang**[1] **Shawn Xu**[1] **Boris Babenko**[1]
**Abbi Ward**[1] **Niels Olson**[2] **Arash Mohtashamian**[3] **Yossi Matias**[1]
**Greg S. Corrado**[1] **Quang Duong**[1] **Dale R. Webster**[1] **Shravya Shetty**[1]
**Daniel Golden**[1] **Yun Liu**[1] **David F. Steiner**[1*] **Ellery Wulczyn**[1*]

## Abstract

Microscopic interpretation of histopathology images underlies many important diagnostic and treatment decisions. While advances in vision–language modeling raise new opportunities for analysis of such images, the gigapixel-scale size of whole slide images (WSIs) introduces unique challenges. Additionally, pathology reports simultaneously highlight key findings from small regions while also aggregating interpretation across multiple slides, often making it difficult to create robust image–text pairs. As such, pathology reports remain a largely untapped source of supervision in computational pathology, with most efforts relying on region-of-interest annotations or self-supervision at the patch-level. In this work, we develop a vision–language model based on the BLIP-2 framework using WSIs paired with curated text from pathology reports. This enables applications utilizing a shared image–text embedding space, such as text or image retrieval for finding cases of interest, as well as integration of the WSI encoder with a frozen large language model (LLM) for WSI-based generative text capabilities such as report generation or AI-in-the-loop interactions. We utilize a de-identified dataset of over 350,000 WSIs and diagnostic text pairs, spanning a wide range of diagnoses, procedure types, and tissue types. We present pathologist evaluation of text generation and text retrieval using WSI embeddings, as well as results for WSI classification and workflow prioritization (slide-level triaging). Model-generated text for WSIs was rated by pathologists as accurate, without clinically significant error or omission, for 78% of WSIs on average. This work demonstrates exciting potential capabilities for language-aligned WSI embeddings.

**Keywords:** histopathology, vision-language model, whole slide image

## 1 Introduction

Recent work in the field of digital histopathology has moved beyond task-specific image classifiers, or even image-only foundation models, to advances using image–text data for vision–language modeling (Huang et al., 2023; Ikezogwo et al., 2024; Lu et al., 2023a; Sun et al., 2024b). The training data for such efforts have predominantly been based on small *patches* or regions-of-interest (ROIs) extracted from within a Whole Slide Image (WSI), paired with associated patch-level text descriptions. For example, the captions and figures for histopathology images in journal articles or educational resources. While such

---

[1] Google Research. Contact: {farukahmed, davesteiner, ewulczyn} AT google.com.
[2] Contributions via Naval Medical Center San Diego, present affiliation Defense Innovation Unit.
[3] Contributions via Naval Medical Center San Diego, present affiliation Renown Regional Medical Center.
[*] These authors jointly supervised the work.

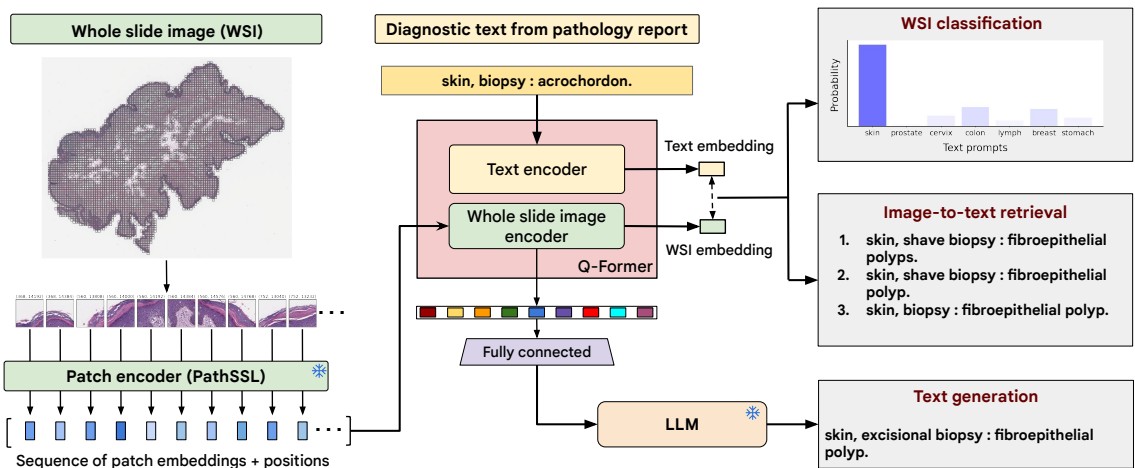

Figure 1: **Model overview.** PATHALIGN provides aligned WSI and text embeddings enabling embedding-based cross-modal retrieval and WSI classification. The WSI-encoder is further aligned with a frozen large language model (LLM), enabling applications such as text generation and visual question answering. The model is trained largely following the BLIP-2 approach (see Section 3.4 for details), making use of a frozen patch-level, histopathology-specialized embedding model (PathSSL) and a frozen LLM.

sources can provide useful pairs for local histological features, many pathology tasks involve slide-level or case-level interpretation. Additionally, curated WSI-level text descriptions accurately paired with specific slides are less readily available than patch-level captions, particularly at the scale necessary for machine learning based approaches. Even when pathology reports are available, it can be challenging to identify the specific slides that are associated with the reported findings. This is because reporting is typically done for the entire case, but there may be many slides for each case, some of which contribute more meaningfully to the diagnosis and reported findings than others. At least in part due to this data-curation challenge, robust strategies to develop visual-language models for WSIs in pathology have been limited to a small number of recent examples.

In this work, we develop PATHALIGN to further address some of the challenges of image–text alignment for gigapixel WSIs (see Figure 1). PATHALIGN learns vision–language alignment using WSIs paired with the corresponding diagnostic text from pathology reports. This approach enables capabilities that rely on image–text alignment at a slide-level, bringing us closer to the possibility of applications such as automatic report generation and case-level visual question answering for digital histopathology. We utilize embeddings from a patch-level foundation model (Lai et al., 2023) as inputs to a BLIP-2 (Li et al., 2023) framework and train two models: one variant trained only with the image–text contrastive loss for efficient embedding-based retrieval, and a second variant using the standard two-stage BLIP-2 training that further integrates a frozen LLM to enable WSI-level text generation and basic visual question-answering capabilities. BLIP-2 provides an effective strategy for pre-training a language-aligned image encoder that can be efficiently fine-tuned for a frozen LLM, and has recently been utilized successfully for report generation with radiology images (Xu et al., 2023). Our results present one of the first quantitative pathologist eval-

uations of WSI-level text generation and image-to-text retrieval across diverse diagnoses. Additionally, we evaluate model performance for WSI classification and present an example of slide prioritization as one practical use case for LLM integration.

## 2 Related work

The emergence of large language models (LLMs) and large multimodal models (LMMs) has created an entirely new field of multimodal and generative AI systems, with approaches such as CLIP (Radford et al., 2021), BLIP-2 (Li et al., 2023), LLaVa (Liu et al., 2024), CoCa (Yu et al., 2022; Zuo et al., 2023), and others. For pathology specifically, a number of recent works describe promising results for vision–language models. These include utilization of a variety of different data sources for image–text pairs, including social media posts (Tsuneki and Kanavati, 2022), YouTube videos and captions (Ikezogwo et al., 2024), pathology reports with patch extraction (Zhang et al., 2023a,b), and large-scale curation of figure-caption pairs from medical literature and educational resources (Lu et al., 2023a; Sun et al., 2024b; Gamper and Rajpoot, 2021; Sun et al., 2024a). Sun et al. (2024a) also recently evaluated many publicly available LMMs on a large, patch-based visual question answering (VQA) dataset, which was itself generated and curated with the help of the GPT-4V LMM. While these works primarily focus on patch-level modeling, initial strategies have also been described to address WSI-level language alignment, with evaluation on a variety of tasks including subtype classification, biomarker status, and report generation (Xu et al., 2024; Shaikovski et al., 2024; Tran et al., 2024). Finally, non-language based approaches towards improved WSI-level modeling have also been explored, including different patch-aggregation strategies (Lu et al., 2023b; Song et al., 2024; Ciga et al., 2021), multimodal pretraining with gene expression data (Jaume et al., 2024), and hierarchical or slide-level self-supervised learning (Chen et al., 2022; Hou et al., 2024).

## 3 Methods

### 3.1 Data

The primary data used for this work consists of a de-identified dataset (DS1) of 354,089 WSIs from a teaching hospital paired with diagnostic text from pathology reports. The vast majority are hematoxylin and eosin (H&E) stained, with a smaller portion of immunohistochemical (IHC) stained slides. DS1 reflects a real-world distribution of case-types for general pathology practice in the U.S. A summary of the most common tissue sample labels is shown in Supplemental Table C.1. The study was reviewed by Advarra IRB (Columbia, Maryland) and deemed exempt from from further review as all data is retrospective and de-identified. In order to further enrich our data for cancer cases, we also included de-identified data from The Cancer Genome Atlas (TCGA). We utilized the set of 12,268 diagnostic WSIs in TCGA across all 32 TCGA solid tumor study types (where each study type approximately maps to a unique cancer type).

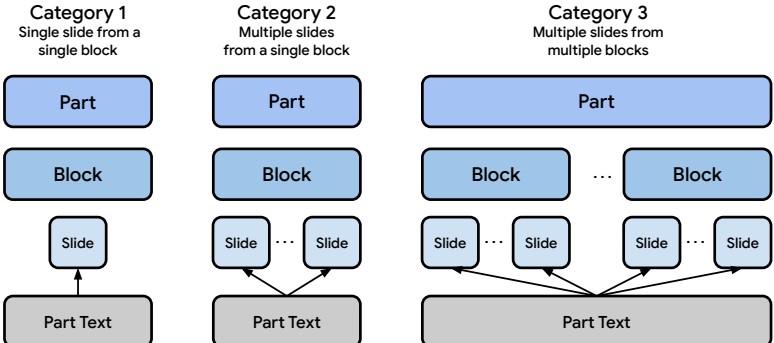

Figure 2: **WSI–text association types in real world data.** We associate each WSI with part-level text from the original report. Due to the *part*, *block*, *slide* hierarchy and variability in accessioning, there are three high-level categories of association between slides and part-level text. The probability that some of the information in the part-level text does not apply to a given slide increases from category 1 to category 3.

### 3.2 Curating image–text pairs

Pathology specimens are typically processed and accessioned by *case*, *part*, and *block*, with findings reported per *part* (where *part* indicates distinct tissue specimens within a single *case*; see also Supplemental Figure B.1). This results in three high-level categories of association between slides and part-level text (see Figure 2): (1) a single slide from a single block; (2) multiple slides from a single block; (3) one or more slides per block across multiple blocks. The probability that some of the information in the part-level text does not actually apply to a given slide (because that particular slide is not representative of the final diagnostic finding) increases from category 1 to category 3. This raises the challenge of pairing any given slide with the portion of the pathology report that actually describes the findings on that particular slide.

To at least partially address this challenge, we first pair each slide with its associated, part-level text from the report using part indicators present in both the full text and the WSI metadata. Next, we separate the DS1 dataset into a "clean" set of all category 1 along with category 2 slides where these ambiguities are mitigated (calling this set DS1-Clean), and a "noisy" set consisting of all categories (DS1-Noisy).

For TCGA, instead of parsing heterogeneous reports to map text to slides without part indicators, we utilized structured case-level metadata available for TCGA (Liu et al., 2018) to generate a basic description in the label : finding format, analogous to the typical structure of part-level text in DS1.

For additional details on creating image–text pairs, see Section A.1 in the Appendix.

### 3.3 Data splits

For DS1, slides from DS1-Clean were randomly split by case into train, validation, and test sets (90/5/5 split). All slides from cases not included in the validation and test splits of DS1-Clean were combined with remaining category 2 and category 3 WSIs to form DS1-Noisy, a larger dataset used for training only ($N = 344,532$ WSIs).

Table 1: **Data overview by case, part, and slide.** Splits were performed at the case-level. DS1-NOISY does not have a validation or test split and the training split of DS1-CLEAN is a subset of DS1-NOISY. TCGA includes only the diagnostic category slides from TCGA. Because DS1-CLEAN has only one WSI per part-level text, the number of slides is equal to the number of parts, but there still may be more than one part per case.

| Name | Description | Split | #Cases | #Parts | #Slides |
|------|-------------|-------|--------|--------|---------|
| DS1-CLEAN | Image–text pairs with only one WSI per part-level diagnostic text. | train | 49,382 | 82,764 | 82,764 |
| | | validation | 2,785 | 4,678 | 4,678 |
| | | test | 2,840 | 4,879 | 4,879 |
| DS1-NOISY | All WSIs with part-level note, including instances of multiple WSIs for the same part-level note. | train | 72,219 | 122,181 | 344,532 |
| | | validation | – | – | – |
| | | test | – | – | – |
| TCGA | All FFPE images in TCGA, with synthesized text based on TCGA study type and metadata. | train | 4,717 | | 6,323 |
| | | validation | 2,172 | N/A | 2,681 |
| | | test | 2,429 | | 3,264 |

For TCGA, diagnostic H&E slides across all TCGA study types ($N = 12,268$) were split into train, validation and test sets on a per-study basis by tissue source site (TSS) to enable better evaluation of generalization across tissue and image processing variability from different sites. TSSs were assigned with a target split-ratio of 2:1:1 across train, validation and test splits within each TCGA study, though the final ratios varied due to site-size variability (see Table 1 and Supplemental Table C.11 for details).

### 3.4 Modeling

**Patch sampling**  We represent each WSI by a set of up to 10,240 tissue-containing patches of size $224 \times 224$ at 10X magnification ($\approx$1 micron-per-pixel), which covers all patches for 97.8% percent of DS1 WSIs, and 91.4% of TCGA WSIs (see supplemental Figure B.3, with additional details in Section A.2.)

**Patch encoder**  We pretrained a pathology-specific patch-level encoder via self-supervised learning using the train split of DS1, following the approach described by Lai et al. (2023) using Masked Siamese Networks (Assran et al., 2022) as the SSL method along with the RandStainNA color augmentation method (Shen et al., 2022). This patch encoder uses a ViT-S architecture (Dosovitskiy et al., 2020; Steiner et al., 2021) and maps $224 \times 224$ pixel patches into embeddings of size 384.

**WSI encoder**  Our WSI-encoder is comprised of the image transformer submodule of the Q-Former in the BLIP-2 framework (Li et al., 2023). The input to the WSI-encoder is the sequence of up to 10,240 patch-embeddings from the patch-encoder, with non-learnable sine and cosine position encodings for the patch coordinates incorporated for both $x$ and $y$ axes (Vaswani et al., 2017). The learned query vectors in the Q-Former are used to cross-attend to the input WSI data.

**Image–text alignment**  PATHALIGN is based on the BLIP-2 (Li et al., 2023) vision–language model architecture and training approach (see Figure 1). In the first stage, the

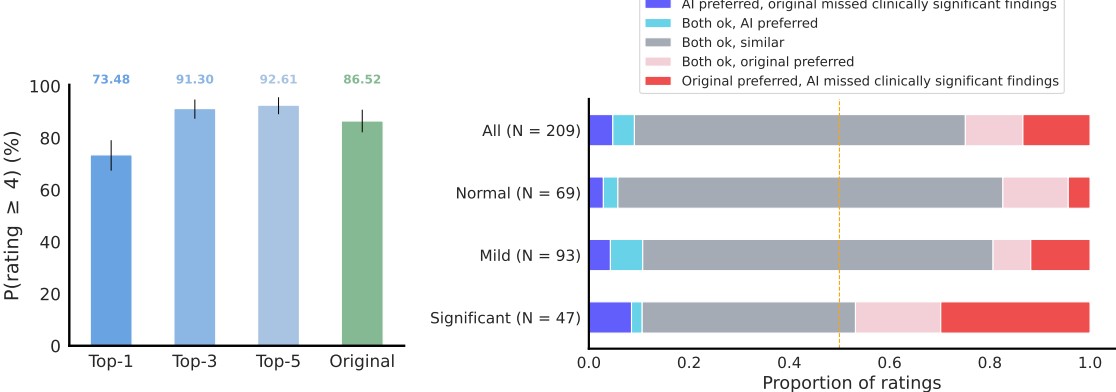

(a) Image-to-text top-K accuracy.     (b) Comparison of generated text with original report.

Figure 3: **Pathologist evaluation of image-to-text retrievals and generated text.** (a) For embedding-based retrievals, top-K accuracy is shown, using a rating of 4 or 5 to indicate accurate text without clinically significant errors or omissions. *Original* refers to pathologist evaluation of the original diagnostic text. (b) Per WSI comparison of ratings for the generated text and the original diagnostic text. Ratings for which both AI and original text received a score of 3 or lower are excluded in this plot: 21 ratings excluded in total (9%) with 3 from *normal* (4% of *normal*), 11 from *mild* (11% of *mild*), and 7 from *significant* (13% of *significant*). The score-based definitions of these categories are provided in Supplemental Table C.9. The *mild* category includes a range of findings such as inflammation, benign conditions, and adenomas. The *significant* category includes carcinoma, dysplasia, and findings with direct implications for clinical management.

WSI and text encoders are trained to align their representations, using learned query vectors to cross-attend to the WSI data. For input to the WSI encoder, WSIs are represented via sequences of patch-level embeddings produced from a patch-level SSL-trained histopathology foundation model along with their positional coordinates. For the second stage, we discard the text encoder from stage 1, and graft the pretrained WSI-encoder to a frozen generative LLM via a linear layer with further fine tuning for text generation. We train one stage 1 model with the image–text contrastive loss only, referring to this variant as PATHALIGN-R (for retrieval, based on use of this model for cross-modal retrieval tasks). The second variant is trained using the standard two-stage BLIP-2 training procedure along with LLM integration for text generation. We refer to this variant as PATHALIGN-G (for generation), and use a frozen PaLM-2 S (Anil et al., 2023) model as the LLM. Additional details are provided in Section A.3 including hyper-parameter settings in Supplemental Table C.3.

### 3.5 Evaluation

**Text retrieval and generation**   Two U.S.-board certified pathologists evaluated texts for top-K image-to-text retrieval (PATHALIGN-R) and text generation (PATHALIGN-G). Automatic evaluation was also performed (primarily for model development) using a similarity score threshold for embeddings from a text-similarity model to determine accurate retrievals (see Section A.5). Text examples were rated on a five-point scale based on concurrent review of the corresponding WSI (scoring instruction details in Supplemental Table C.8). Retrieval

Table 2: **Examples of retrieved and generated text for input WSIs.** These qualitative examples illustrate the style for evaluated text-image pairs and highlight the model's ability to retrieve and generate accurate text, sometimes even preferred to the original diagnostic text.

| | Example 1 | Example 2 | Example 3 |
|---|---|---|---|
| WSI thumbnail |  |  |  |
| Enlarged view |  |  |  |
| Original text | duodenum, biopsy : unremarkable intestinal mucosa. | cervix : biopsy: - low grade squamous intraepithelial lesion (cin 1, mild dysplasia). | skin, biopsy : intradermal nevus. |
| Top retrieved text | duodenum, third part, biopsy : small bowel mucosa with no pathologic diagnosis. | cervix : biopsy: - high grade squamous intraepithelial lesion (cin-2; hsil). | skin, punch biopsy : intradermal nevus. |
| Generated text | duodenum, biopsy : duodenal mucosa with no significant pathologic changes. | cervix, biopsy : low grade squamous intraepithelial lesion (cin 1). | skin, punch biopsy : compound nevus. |
| Pathologist review | Agree with all | Favor HSIL (high grade) | Favor compound nevus |

was performed using cosine-similarity between image and text embeddings using the corpus of unique texts in the test set ($N = 3,176$ unique diagnostic texts). Additional details including information about the retrieval task setup and the 120 test set WSIs sampled for pathologist evaluation are provided in Section A.4.

**WSI classification** We evaluated PATHALIGN-R on four WSI classification tasks: (1) NSCLC subtyping: non-small cell lung cancer subtyping using LUAD and LUSC in TCGA; (2) RCC subtyping: renal cell carcinoma subtyping using KIRC, KIRP and KICH in TCGA; (3) BRCA subtyping: breast cancer subtyping of ductal versus lobular carcinoma using BRCA and subtype metadata in TCGA; (4) Procedure type: biopsy vs. resection classification using a subset of DS1. To perform classification, WSI embeddings are compared to text embeddings for the classes of interest, using a curated set of texts per class. Texts used for each class are provided in Supplemental Table C.7. See Section A.4.3 for additional WSI classification details. Confidence-intervals were computed via bootstrapped resampling with replacement over 1000 replicates.

## 4 Results

**Image-to-text retrieval** Pathologist evaluation of image-to-text retrieval is summarized in Figure 3a. Top-1 and top-3 retrieval accuracy were 73.5% and 91.3%, respectively (based on a rating score of 4 or 5 to define accurate text). The original diagnostic text was scored as 4 or 5 for 86.5% of ratings. Plots for the individual raters are provided in Supplemental Figure B.4a. Sub-analysis by "common" and "less common" specimen-type categories

Table 3: **WSI classification.** Classification results for NSCLC, RCC, and BRCA subtyping (TCGA), and procedure type (DS1). See Table C.7 for text-prompts corresponding to each class. 95% confidence-intervals were computed via bootstrapped resampling with replacement over 1000 replicates.

| Task | AUC | Balanced accuracy |
|---|---|---|
| NSCLC Subtyping | $0.945 \; [0.921 - 0.965]$ | $0.875 \; [0.838 - 0.909]$ |
| RCC Subtyping | $0.971 \; [0.954 - 0.985]$ | $0.889 \; [0.832 - 0.941]$ |
| BRCA Subtyping | $0.879 \; [0.823 - 0.926]$ | $0.775 \; [0.706 - 0.836]$ |
| Procedure Classification | $0.987 \; [0.976 - 0.996]$ | $0.942 \; [0.912 - 0.978]$ |

did not suggest bias towards retrieval of common cases (Supplemental Figure B.7). While automatic evaluation of image-to-text retrieval (as well as text-to-image and image-to-image retrieval) was also performed, this was primarily used for hyper-parameter tuning; details and test set results are in Section A.5 and Supplemental Table C.4.

**Image-based text generation** Evaluation of generated text is summarized in Figure 3b and Figure B.6, with examples in Table 2. For images where either original text or AI generated text was rated 4 or above, the AI generated text was determined to be equivalent or better than the original text in 75% of ratings. For all WSIs ($N = 115$ images), generated text was rated to be 4 or 5 (*i.e.* mostly or highly accurate) for 78% of ratings. See Figure 3b and Figure B.6 for complete results, including subanalysis by finding type of normal, mild, or significant (as based on the original diagnostic text). Data for the individual raters as well as the inter-rater confusion matrix for scoring of original diagnostic texts are provided in Supplemental Figure B.4b and Supplemental Figure B.5, respectively.

**WSI classification** Results for LUAD, RCC, BRCA, and procedure type classification are summarized in Table 3. Texts for each class are provided in Supplemental Table C.7.

**Exploring additional vision–language applications** To highlight one potential application utilizing LLM integration, we demonstrate a case prioritization example. We randomly select 200 colon biopsies representing a theoretical case load and use PathAlign-G to return the set, sorted by likely "severity" of the findings. The results and prompt are summarized in Supplemental Table C.10. While not perfect, all carcinoma cases are appropriately in the top category, most tubular adenomas and other findings in the second category, and most hyperplastic polyps along with benign biopsies in the lowest risk category, thus highlighting the promising potential to organize or group cases with flexible natural language queries.

## 5 Conclusion

This work demonstrates the novel development of multimodal pathology models using WSIs paired with curated portions of original diagnostic reports along with a pre-trained patch encoder and a LLM. These initial results highlight the potential for WSI-text alignment in a manner that can incorporate the reasoning capabilities of large multimodal models. We provide additional discussion in Section A.6 in the Appendix.

## Acknowledgements

We thank Wei-Hung Weng, Tiam Jaroensri, and Michael Howell for useful feedback on the manuscript. We thank the Google Research team for software and hardware infrastructure support as well as operations team members involved in the digitization and program management aspects related to this study; especially Melissa Moran, Robert MacDonald, Allen Chai, Robert Nagle, and Josh Pomorski. We also thank Kenneth Philbrick, Liron Yatziv, Can Kirmizi, and Rory Pilgrim for helpful technical discussions. We acknowledge James Wren and Colin Wageman for data-related discussions and thank the pathologists who reviewed model output for this study. We thank Todd Lilje, Daniel Ward, and the Naval Medical Center San Diego Laboratory and Clinical Investigations Departments for administrative and research support and guidance. N.O. is a military Service member. This work was prepared as part of their official duties. Title 17, U.S.C., §105 provides that copyright protection under this title is not available for any work of the U.S. Government. Title 17, U.S.C., §101 defines a U.S. Government work as a work prepared by a military Service member or employee of the U.S. Government as part of that person's official duties. The study protocol was approved by the Naval Medical Center San Diego Institutional Review Board in compliance with all applicable Federal regulations governing the protection of human subjects. Support for this study included funding support from Google under NCRADA-16-471. The results shown here are in part based upon data generated by the TCGA Research Network: https://www.cancer.gov/tcga.

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

## Appendix A. Supplemental methods

### A.1 Creating image–text pairs

As is typical for pathology reports, each case in DS1 has an associated diagnostic text, corresponding to the final diagnosis (*i.e.* "bottom line" text) from the pathology report (see Supplemental Figure B.2). These diagnostic texts are structured into part-level text sections. Because each case may have several different parts, we parse these to the part-level via regular expressions. For each part-level text section, there is a *label* (description of tissue site or surgical procedure) and a *finding* (description of diagnostic findings). Because the *label* text is typically based on the specimen processing and preparation, it often includes information such as anatomic location and laterality that may not be inferable from the WSI alone. Occasionally the findings section also includes this type of information. As such, we further apply a set of regular expressions to remove information from both the labels and findings that cannot reliably be determined from the WSI. Additionally, as diagnostic reporting in pathology often exhibits a common style across pathologists, many free text descriptions for different slides will be the same when the findings are essentially the same.

Within the framework of the three categories established for part-to-slide mappings, it is also possible that the pathologist reviewed more slides than those that were archived and subsequently digitized, so even category 1 has some possibility of the text not corresponding specifically to the image. While we believe this possibility to be a rare occurrence in DS1, complete accessioning metadata from each case was not available, so formal quantification of "missing slides" could not be performed.

For TCGA, although pathology reports are available (as PDFs), they are submitted from a variety of source sites with significant variability in structure and detail. Additionally, they do not specify which portion of the report corresponds to the available images. Instead of parsing these heterogeneous reports, we utilized structured case-level metadata available for TCGA (Liu et al., 2018) to generate a basic description in the `label :   finding` format, analogous to the typical structure of part-level text in DS1. Specifically, we used the tissue type, histological type, and histologic grade (when available) to generate captions such as `bladder, resection :   histological type: invasive urothelial carcinoma; tumor grade:   high grade`. We associate each slide with a caption generated from the metadata in this way. For TCGA, the possibility that a given slide in isolation is not representative of this metadata is at least partially mitigated because typically only one or two diagnostic slides per case were submitted and these slides were selected to be representative of the case-level diagnosis and to have substantial tumor content.

### A.2 Patch sampling

Tissue masks were generated via a sequence of image processing operations. These consist of transforming RGB images to HSV-space, performing morphological operations to group together connected regions, thresholding on pixel intensity, and post-processing with an erosion operation to remove remaining noise. Using these tissue masks, we identify all tissue containing patches of size $224 \times 224$ pixels, with a stride of 192 pixels (32 pixels of

overlap). We use at most 10,240 patches per WSI (sampling without replacement when the total number of tissue-patches exceeds this count).

## A.3 Modeling: image–text alignment

To avoid false negatives from similar diagnostic findings within training batches in the image–text contrastive (ITC) loss and image–text matching (ITM) loss, we mask out negative pairs, $(\text{image}_i, \text{text}_j)$, $i \neq j$, with high text similarity between $\text{text}_i$ and $\text{text}_j$ using text embeddings from the Universal Sentence Encoder model (Cer et al., 2018). We threshold on a cosine similarity of 0.985, which typically requires very high similarity with only slight differences in syntax or word choice (see Supplemental Table C.2 for examples). To further reduce the impact of false-negatives, we did not use hard-negative mining for the ITM loss.

For PATHALIGN-R, we chose not to use ITM reranking during retrieval for practical considerations, with efficiency in terms of potential model-serving and typical client-end API use in mind. When using the cosine similarity between contrastive embeddings for ranking, we found that training with the ITC loss alone and using a single learnable query in the Q-Former worked better for retrieval. For text generation (PATHALIGN-G), we found that the standard BLIP-2 approach of training a stage 1 model including all losses worked best, along with 32 learnable query vectors. Hyper-parameter settings are provided in Supplemental Table C.3. All model selection choices were made using the validation set.

## A.4 Evaluation details

### A.4.1 PATHOLOGIST EVALUATION

A subset of 120 test set WSIs from the DS1 test set was selected for evaluation by pathologists. For subset selection the test set was first divided into two categories: (1) the most common specimen types (colon, rectum, cervix, and skin biopsies) and (2) other specimen types. Then, 60 images from each of these two categories were sampled. For each WSI, pathologists were presented with the image in a web-based digital pathology viewer along with five retrieved diagnostic texts (from PATHALIGN-R), a generated text (from PATHALIGN-G), and the original diagnostic text. Texts were provided in random order and pathologists were blinded to the source of each (*i.e.* retrieved, generated, original) to avoid any bias in interpretation. At evaluation, 5 images were dropped due to pathologist review indicating poor image quality ($N = 1$) or need for immunohistochemistry (IHC) ($N = 4$) for confident interpretation.

### A.4.2 EVALUATING IMAGE-TO-TEXT RETRIEVAL

For image-to-text retrieval, PATHALIGN-R takes a WSI and corpus of texts as input, and scores the texts according to embedding similarity with that of the input image. The text corpus is comprised of all unique diagnostic texts from WSI-text pairs in the DS1 test set ($N = 3,176$ unique diagnostic texts). Similarity scoring was performed using cosine-similarity between embeddings for the input WSI and the diagnostic text. For measuring top-K retrieval accuracy, we consider a retrieved text as being accurate if it received a rating of at least 4 (*i.e.* mostly accurate without clinically significant error or omission). To address the fact that retrieval results could be influenced by the frequency of similar

cases in the corpus, we limited retrieval to unique diagnostic texts (*i.e.* duplicates removed before retrieval), and we also performed sub-analysis on the "common" and "less common" specimen types as defined in prior sections.

### A.4.3 WSI CLASSIFICATION

To perform classification, PATHALIGN-R is given a WSI and a class is assigned based on similarity of the image embeddings to the text embeddings for classes of interest, using a curated set of texts per class. This can be thought of as a highly constrained version of image-to-text retrieval, except that instead of scoring the similarity between the model's WSI embedding and embeddings for the corpus of texts, the scoring is done using the average similarity between the WSI embeddings and the set of texts associated with each class. This approach is often referred to as zero-shot classification in the literature, but since the concepts in these tasks are contained in the training data, we refer to this simply as WSI classification here.

For the three subtyping tasks, texts for each class were curated to represent common diagnostic texts for WSIs of each subtype in the training set (identified via regular expression matching). For the procedure classification task, the biopsy class is represented by just the single word `biopsy`, while the resection class is represented by `resection` as well as a variety of tissue specific resection types (*e.g.* `lobectomy`, `mastectomy`, `nephrectomy`, etc; see Supplemental Table C.7). WSIs for the procedure classification task were selected by randomly sampling 250 WSIs with a diagnostic text containing the word `biopsy` and selecting all WSIs containing any of the resection texts ($N = 63$) from the DS1 test set. We evaluate classification performance with both macro-averaged AUC (using average similarity directly) as well as balanced accuracy (taking the max similarity score across classes). Confidence-intervals were computed via bootstrapped resampling with replacement over 1000 replicates.

### A.5 Automatic evaluation

During model development, we used automatic evaluations for cross-modal retrieval and diagnostic text generation to guide hyper-parameter selection and modeling choices. The methods we used are described below and test set results for the final models are reported below.

**Cross-modal retrieval** We performed cross-modal retrieval analysis for image–text pairs at the WSI-level, a task with implications for finding and curating cases of interest across educational, research, and clinical workflows.

In the image-to-text retrieval setting, the model is given a WSI and tasked with scoring a corpus of diagnostic texts according to how relevant they are for the given WSI. In our case, the text corpus consists of all unique diagnostic texts from WSI-text pairs in the validation dataset. The scoring is done using cosine-similarity between the model's embedding for the input WSI and the model's embeddings for all diagnostic texts in the corpus. Because there may be many texts that accurately match the image, a key challenge in evaluating this type of retrieval is determining the full set of diagnostic texts in the corpus that match with the input WSI, which is required for standard retrieval metrics such as MAP, NDCG and top-K accuracy.

Since our datasets consist of [WSI, diagnostic text] pairs, we have one known ground-truth diagnostic text match. However, there are potentially many other texts in the corpus that describe the same diagnostic finding in different ways. To estimate the set of matching diagnostic texts we compute the cosine similarity between embeddings from the ground-truth text and all other texts in the corpus using the Universal Sentence Encoder model (Cer et al., 2018). Any text with a cosine similarity score above a threshold of 0.985 was considered a match. This threshold was manually tuned to be as low as possible without including false positive matches (on the validation set). However, due to this high similarity threshold and failures in the Universal Sentence Encoder to map syntactically different yet diagnostically equivalent texts to similar embeddings, not all true-positive matches are included (see examples in Table C.6). This is a limitation of this automated, yet, large-scale retrieval analysis, addressed through evaluations performed by a human expert reviewing both images and text.

In the text-to-image retrieval setting, the model is given a diagnostic text and tasked with scoring a corpus of WSIs according to how relevant they are to the diagnostic text. The retrieval analysis was performed analogously to the image-to-text setting, with the set of matching WSIs estimated using similarity between the input diagnostic text and the diagnostic texts associated with each WSI in the corpus of WSIs. Results are summarized in Table C.4.

**Image-to-image retrieval** We also evaluated image-to-image retrieval, i.e. the problem of finding images with similar associated diagnostic text. This was done analogously to cross-modal retrieval except that the input image is excluded from the set of matching images and input images that do not have any matching images (i.e. there are no other images in the image corpus the where similarity between associated texts is above the required threshold) were excluded from analysis. For automatic evaluation of image-to-image retrieval, the similarity score between the original texts for the input and the retrieved images were calculated, again using a threshold of 0.985 for defining accurate retrieval. Results are summarized in Table C.6, where they are presented in comparison to performance using the averaged patch-level embeddings from the domain-specific patch encoder (PathSSL) that we used to embed patches for our model. Text generation Text generation was evaluated automatically by computing ROUGE-L (Lin, 2004) and METEOR (Banerjee and Lavie, 2005) scores between original and generated diagnostic text. Results are summarized in Table C.5.

## A.6 Discussion

Many important applications in histopathology involve interpretation of WSIs. Leveraging advances in efficient vision–language pretraining (Li et al., 2023) and self-supervised patch-level encoders (Lai et al., 2023), we develop a pathology report aligned WSI-encoder using a real-world dataset of over 350,000 gigapixel WSIs with diagnostic text from associated pathology reports. We evaluate this model for classification, cross-modal retrieval, and generation of text describing pathologic findings.

Our work complements recent efforts on language aligned WSI-encoders such as PathM3 (Zhou et al., 2024), PRISM (Shaikovski et al., 2024) and GigaPath (Xu et al., 2024). Compared to prior work, we explore an alternative method for efficient image–text alignment

with WSIs based on the BLIP-2 approach. This enables us to align our WSI-encoder with a pre-trained LLM (PaLM-2 S) for generating text from WSIs. While evaluation of the other recent models has been limited to classification tasks,automated scoring, and qualitative review of text generation, we report the first quantitative pathologist evaluation of cross-modal retrieval and text generation.

The text generation evaluations provide several interesting insights. On one hand, they reflect the impressive capability of the domain-specific WSI-encoder to align with a pre-trained LLM even when images are gigapixel-sized. The generated texts are generally quite accurate at reflecting important information about the WSI, often showcasing important slide-level capabilities by providing information that requires aggregating information and context from multiple patches. Examples of this include the type of procedure or biopsy, and perhaps more impressively, the concept of low grade versus high grade cervical dysplasia, which is defined in part by the extent of the epithelial thickness that is affected, and thus likely requires contextual information within the slide (example in Table 2).

On the other hand, there are clearly still some shortcomings in the details provided by the generated text, such as specific grades for prostate and breast cancer. We also observe some confabulations, particularly in the specimen type when specimen information cannot be readily inferred from the image (*e.g.* neck contents, lymph node). This is likely due, in part, to imperfect removal of this type of specimen information when we processed part labels, but also reflects the inherent importance of context when reviewing slides and writing reports. While prompting the model with the specimen information along with the image is one strategy that might reflect real world availability of the part label during slide review, we did not find this to significantly improve text generation in our study. Efforts to more effectively clean training data and to more thoroughly evaluate confabulations and optimize prompting strategies using available metadata are opportunities for future work.

While PATHALIGN-R performs well on the cancer subtyping tasks (see Table 3), direct comparison to other image–text pathology models (*e.g.* CONCH (Lu et al., 2023a), Giga-Path (Xu et al., 2024)) cannot be made directly due to different image splits, as well as our inclusion of TCGA data in training (albeit from different TSS than those used for testing).

While our training datasets are comparable in size to prior work on WSI-level image–text alignment (Xu et al., 2024; Shaikovski et al., 2024), they are relatively small compared to datasets that have been used for image–text alignment from natural images ($> 400M$). In initial experiments, we observed that training on the full set of training data (DS1-NOISY) provided significantly better performance than training on only the cleaner, but smaller train split of DS1-CLEAN, further supporting the potential for data scaling. Because pathology reports are in principle available for all WSIs that have gone through clinical workflows, we hope to see future work build on our approach with larger-scale datasets of WSIs paired with pathology reports.

### A.6.1 LIMITATIONS

This work has several limitations. A primary challenge in aligning pathology WSIs with diagnostic text is the *many-to-one* nature of slides to the associated portion of the diagnostic report. In this work, we curated our dataset in a manner that minimized this issue for the validation and test splits, but this also resulted in a relative enrichment in these splits for

the types of cases that typically only have one slide per submitted pathology *part*, such as colon, skin, and cervical biopsies along with other small specimens. Additionally, there are at least some instances for which there are "missing" slides, such that the single slide used for inference may not contain all the information represented in the paired text. Curation of datasets to include only the representative slides for the reported findings and modeling at the level of multiple slides remain as opportunities to further address this issue.

The use of TCGA, while useful for enriching DS1 with cancer cases and providing increased training diversity, also introduces specific limitations. For example, we used available structured metadata to generate "synthetic" diagnostic captions for these images. While an effort was made to diversify the language used for describing any given entity, these captions still do not necessarily represent the reporting style for these types of cancer cases in real clinical reports. For example, an actual diagnostic report for a cancer resection might describe many aspects of the tumor grading, staging, and subtyping in a manner that is more extensive than the available structured metadata from TCGA.

Due to lack of available out-of-distribution datasets for this study, image-to-text retrieval and text generation tasks were only evaluated on in-distribution data for DS1. Experiments to evaluate generalization of this model to diverse data sources are warranted. Analysis on additional tasks such as text-to-image retrieval could also be performed and existing evaluations could be improved by increasing the evaluation set size, including greater diversity of cases and findings, and including a larger number of raters.

### A.6.2 Future work

Future work could explore additional vision–language modeling strategies coupled with different LLMs and further instruction tuning. While we benefited from the lower computational costs by using relatively fewer number of query vectors, direct patch-to-patch interaction across the WSI is potentially not fully captured in the cross-attention mechanism and might be improved further, such as through efficient self-attention. Modeling at the level of multiple slides across entire parts or cases, higher magnifications, or a pyramid of multiple magnifications could also further enable useful applications.

## Appendix B. Supplementary figures

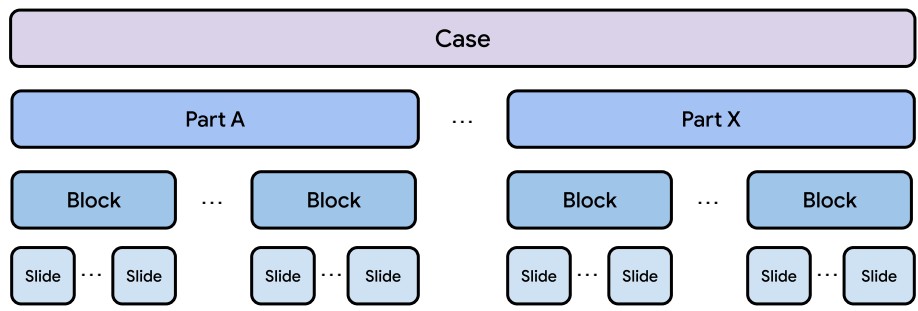

Figure B.1: **Overview of pathology case accessioning.** Pathology specimens are typically processed and accessioned by case, part, block, and slide. A single case may have several different parts and a single part may have several different blocks, with each block sectioned (*i.e.* cut) to provide one or more slides for histopathology review.

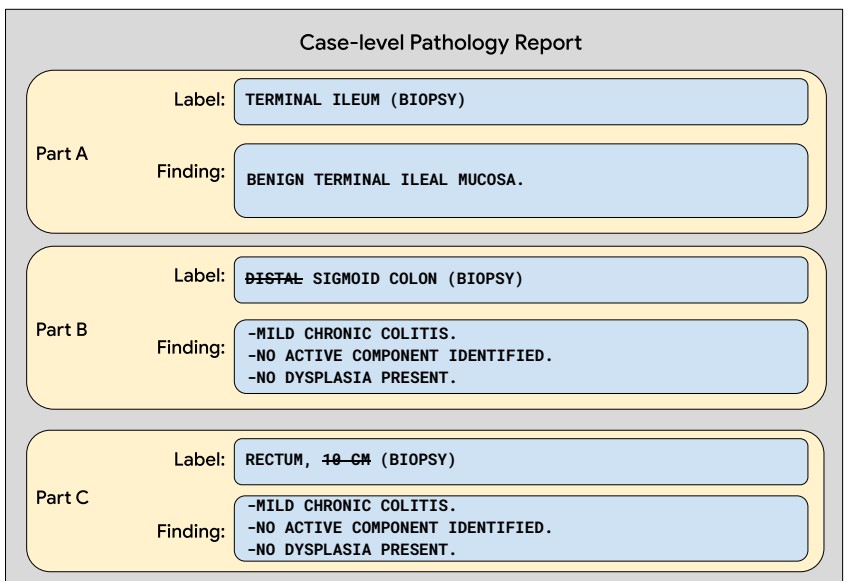

Figure B.2: **Example part-level diagnostic text from DS1.** An example of final diagnosis text from a pathology report for a colorectal biopsy case, with reports split by part. Information that may not be determined from the images (e.g. sample location, tumor size) is removed on a best effort basis via regular expressions, with example removals indicated by strikethrough text in this figure.

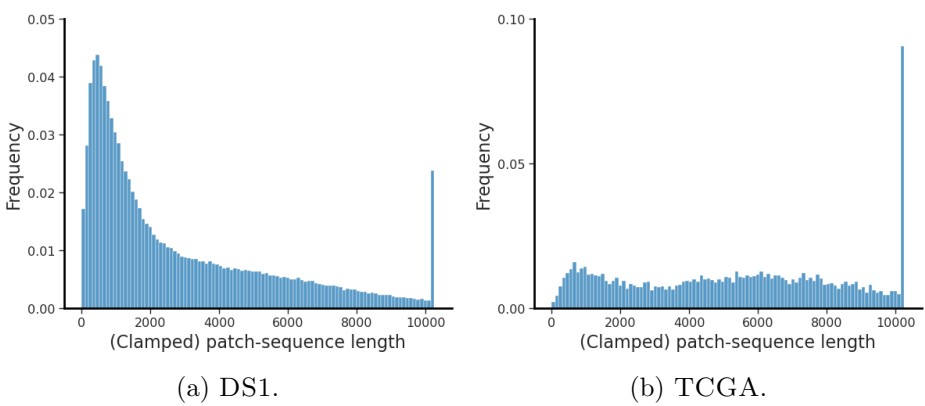

(a) DS1.

(b) TCGA.

Figure B.3: **Histograms for number of patches sampled per WSI.** Up to 10,240 tissue-containing patches were sampled per WSI (without replacement), which covers all tissue-containing patches for (a) 97.8% of DS1 WSIs, and (b) 91.4% of TCGA WSIs across all splits.

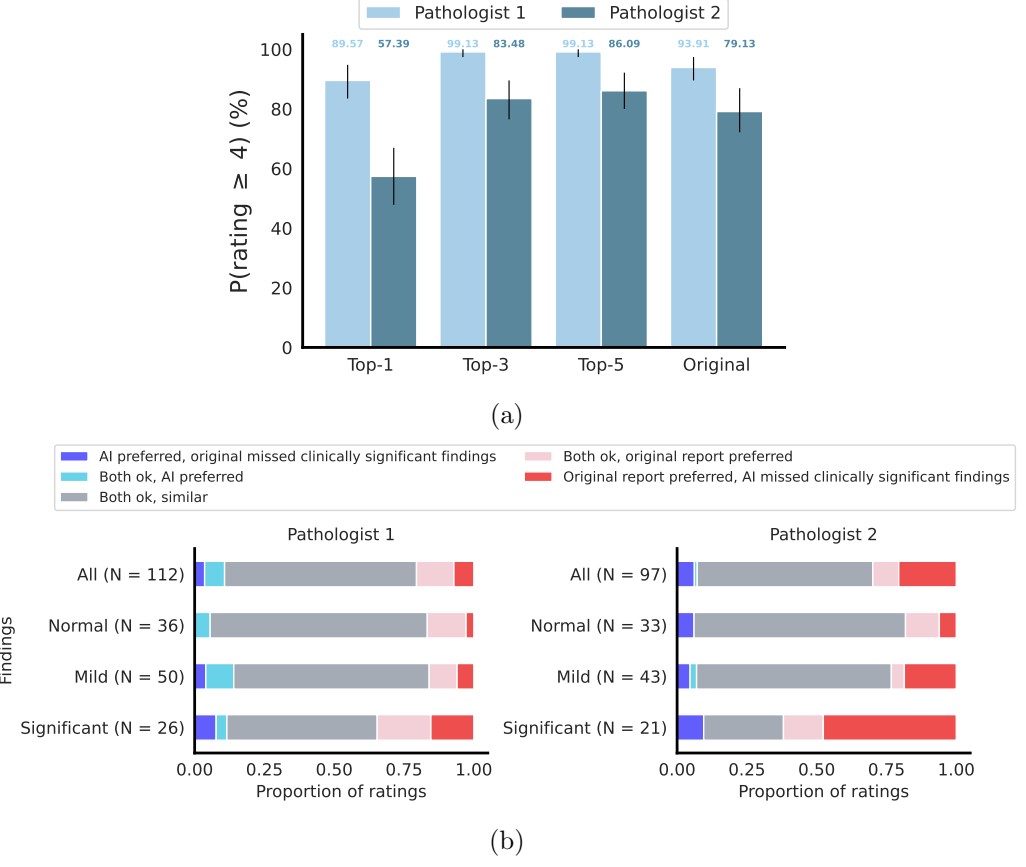

Figure B.4: **Pathologist evaluation by individual rater.** Results for (a) retrieval and (b) generated text are shown for each rater, corresponding to Figure 3a and Figure 3b of the main text, respectively. The general trends for each rater are similar with respect to the original diagnostic text, suggesting the inter-rater variability is at least partially explained by "calibration differences" in regards to interpretation of the scoring system.

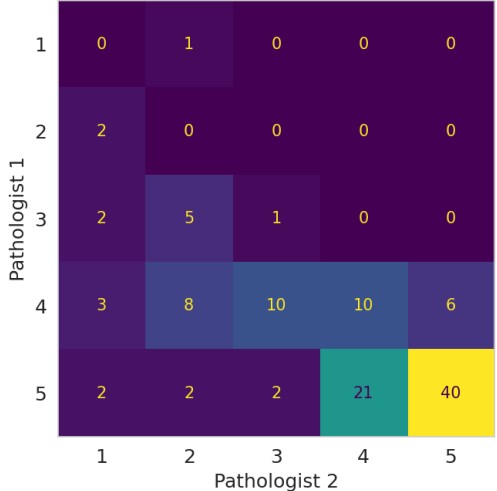

Figure B.5: **Inter-rater comparison for scoring of original diagnostic text** A confusion matrix for pathologist scoring of the original diagnostic text to offer some additional perspective on inter-rater agreement and scoring calibration.

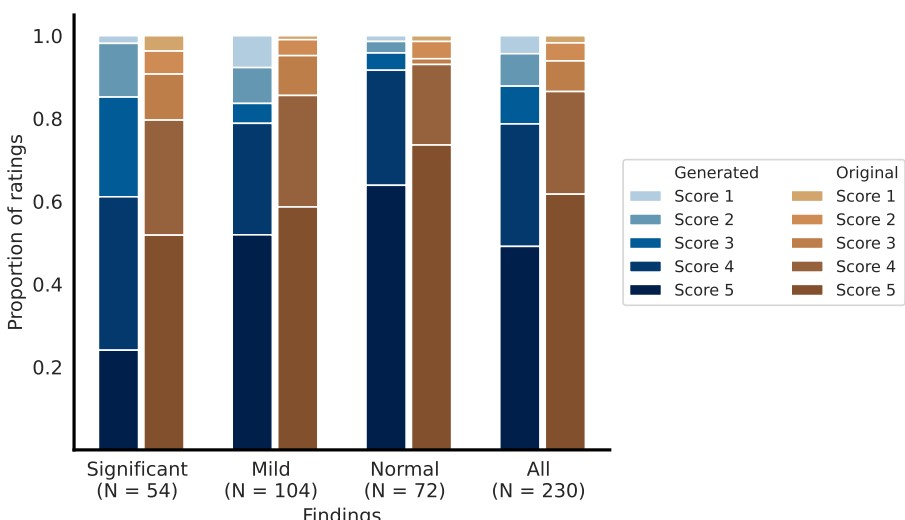

Figure B.6: **Complete scoring results by category for generated text and original diagnostic text.** For images across each finding category, the portion of ratings corresponding to each possible score is plotted for the generated text and original text, respectively. The sum of ratings 4 and 5 gives the portion of ratings indicating *mostly* or *highly* accurate text.

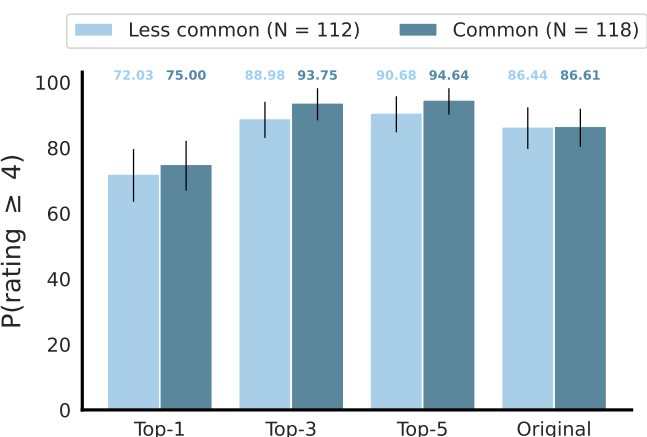

Figure B.7: **Retrieval performance for common and less common specimen types.** To at least partially address the potential for retrieval results to be influenced by the frequency of similar cases in the corpus, we also performed subanalysis by the "common" and "less common" specimen categories (as defined in Section A.4.1). Plotted data represents the portion of WSIs for which at least one of the top-K retrieved texts was scored as 4 or 5 upon pathologist review, averaged across two pathologists. Counts for number of ratings corresponding to each category are provided. Error bars are 95% confidence intervals via bootstrapping over WSIs.

## Appendix C. Supplementary tables

Table C.1: **Top part label frequencies in DS1.** Top 25 most common part labels and their frequency in DS1 (from $N = 122,181$ total diagnostic texts).

| Part Label | Percentage | Part Label | Percentage |
|---|---|---|---|
| colon, biopsy | 6.19 | endocervical curettings | 0.94 |
| skin, shave biopsy | 5.49 | cervix, leep | 0.91 |
| skin, excisional biopsy | 4.17 | breast, mastectomy | 0.83 |
| cervix, biopsy | 3.72 | prostate, prostatectomy | 0.83 |
| lymph node, excision | 2.43 | breast, lumpectomy | 0.83 |
| skin, punch biopsy | 2.3 | duodenum, biopsy | 0.78 |
| cervix | 2.2 | placenta | 0.74 |
| cervical biopsy | 1.77 | appendix, appendectomy | 0.57 |
| rectum, biopsy | 1.48 | endometrial biopsy | 0.56 |
| skin, biopsy | 1.41 | gallbladder, cholecystectomy | 0.56 |
| colon, polypectomy | 1.12 | endocervical curettage | 0.53 |
| esophagus, biopsy | 1.06 | prostate, biopsy | 0.51 |
| stomach, biopsy | 0.98 | | |

Table C.2: **Qualitative examples of the automated text similarity score.** Similarity scores as computed by embeddings from the Universal Sentence Encoder, for a set of selected examples with scores above and below our conservative threshold of 0.985. Scores for which the threshold correctly identifies equivalent or different pairs, respectively, are color coded with green for correct and red for incorrect.

| Threshold | Text $i$ | Text $j$ | Score$_{ij}$ |
|---|---|---|---|
| < 0.985 | soft tissue, supraclavicular region, biopsy : classical hodgkin lymphoma. | cervix : biopsy: high grade squamous intraepithelial lesion (cin-ii). | 0.619 |
| | colon, cecum, polypectomy : fragments of tubulovillous adenoma. | colon, polypectomy : tubular adenoma. | 0.681 |
| | cervical polyp biopsy : low-grade squamous intraepithelial lesion (cin i). | cervix, biopsy : benign squamous mucosa, no transformation zone identified. | 0.771 |
| | skin, excisional biopsy : neurofibroma. | skin, excisional biopsy : dermatofibroma. | 0.953 |
| | colon, biopsy : hyperplastic polyp. | colon, biopsy : adenomatous polyp. | 0.957 |
| | cervix, biopsy : high-grade squamous intraepithelial lesion. | cervix, biopsy : low-grade squamous intraepithelial lesion. | 0.970 |
| | uterine cervix, biopsy : benign cervical tissue, no dysplasia identified. | cervix : biopsy: benign cervical tissue, no dysplasia identified. | 0.978 |
| > 0.985 | skin, excisional biopsy : epidermal inclusion cyst, excised. | skin, excisional biopsy : epidermal inclusion cyst. | 0.990 |
| | skin, shave biopsy : ulcerated sclerosing basal cell carcinoma; extends to the base of the biopsy. | skin, shave biopsy : basal cell carcinoma, extends to the base of the biopsy. | 0.989 |
| | cervix, biopsy : low grade squamous intraepithelial lesion (cin-i). | cervix : biopsy: low grade squamous intraepithelial lesion. | 0.989 |
| | colon, biopsy : chronic active colitis, no dysplasia identified. | colon, biopsy : chronic active colitis, mild. -no dysplasia identified. | 0.987 |
| | appendix, appendectomy : acute suppurative appendicitis. | vermiform appendix, appendectomy : acute suppurative appendicitis. | 0.987 |
| | colon polyp, biopsy : adenomatous polyp. | colon, polyp, excisional biopsy : adenomatous polyp. | 0.987 |
| | terminal ileum, biopsy : small bowel mucosa with no pathologic diagnosis. | ileum, terminal, biopsy : small bowel mucosa with no pathologic diagnosis. | 0.986 |

Table C.3: **Model hyper-parameters.** Hyper-parameter tuning and checkpoint selection for our retrieval and classification model (PATHALIGN-R) were done using automatic evaluation of retrieval tasks (average of top-1 retrieval accuracy, NDCG, and average precision) on the validation set. For PATHALIGN-G, we used ROUGE-L recall and similarity scores from the Universal Sentence Encoder model, along with visual inspection of captioning quality, to guide hyper-parameter tuning and checkpoint selection.

| Shared: Stage 1 | |
| --- | --- |
| Initialization | Random |
| ITC/ITM false-negative masking | Yes |
| Q-Former query dimension | 192 |
| Q-Former intermediate dimension | 3072 |
| ITC projection layer dimension | 128 |
| Learning rate scheduler | Linear warmup + cosine decay |
| Learning rate | $1e-4$ |
| Weight decay | 0.05 |
| AdamW $\beta_1, \beta_2$ | 0.9, 0.998 |
| Linear-warmup steps | 2000 |
| Maximum training steps (w/ early stopping) | 100000 |
| Batch-size | 1024 |
| Learnable constrastive temperature | Yes |
| **PATHALIGN-R: Stage 1** | |
| Learnable queries | 1 |
| ITC, ITM, ITG loss coefficients | 1.0, 0.0, 0.0 |
| Initial contrastive temperature | 0.01 |
| **PATHALIGN-G: Stage 1** | |
| Number of learnable query vectors | 32 |
| ITC, ITM, ITG loss-coefficients | 1.0, 0.5, 1.0 |
| ITM false-negative masking | Yes |
| Learning rate | $1e-4$ |
| Initial contrastive temperature | 0.07 |
| **PATHALIGN-G: Stage 2** | |
| Optimizer | Adafactor with Adam |
| Learning rate | $5e-5$ |
| Adam $\beta_1, \beta_2$ | 0.9, 0.999 |
| Warmup steps | 1000 |
| Weight decay | $1e-10$ |
| Maximum training steps (w/ early stopping) | 200000 |
| Batch size | 64 |
| Gradient-clipping norm | 10.0 |
| LLM | PaLM-2 S |
| Decoding | greedy |

Table C.4: **Test set cross-modal retrieval results for PathAlign-R using automatic evaluation.** In all cases, the text associated with the input or retrieved images was used to determine match, with a text similarity score threshold of 0.985 using the Universal Sentence Encoder model. These metrics (for validation sets) were primarily used for model development, with pathologist evaluation providing a more interpretable analysis (see Figure 3a); but the test set automatic evaluation results are provided here for reference. The relatively low number of unique TCGA texts is due to the nature of synthesizing TCGA text from metadata as descried in the Methods. For examples of similarity score pairs, see Table C.2. MAP: Mean Average Precision; NDCG: Normalized Discounted Cumulative Gain.

| Dataset | Query | Corpus | MAP | NDCG | Top-1 | Top-5 | Top-10 |
|---------|-------|--------|-----|------|-------|-------|--------|
| DS1 | Text | Image ($N = 4,876$) | 0.22 | 0.43 | 0.16 | 0.41 | 0.57 |
| | Image | Text ($N = 3,177$) | 0.21 | 0.37 | 0.10 | 0.33 | 0.48 |
| TCGA | Text | Image ($N = 3,264$) | 0.49 | 0.76 | 0.58 | 0.87 | 0.9 |
| | Image | Text ($N = 161$) | 0.50 | 0.62 | 0.34 | 0.73 | 0.84 |

Table C.5: **Automatic evaluation results for generated text**. Values reported are the computed metrics as indicated for generated text compared to original diagnostic text over all test set images. Since diagnostic texts for TCGA were synthesized from metadata across splits, we do not evaluate text generation for TCGA.

| Dataset | ROUGE-L (F-Measure) | METEOR |
|---------|---------------------|--------|
| DS1 ($N = 4,876$) | 0.579 | 0.612 |

Table C.6: **Automatic evaluation for image-to-image retrieval.** Test set retrieval metrics for image-to-image retrieval using the contrastive WSI-embedding from PathAlign-R. Embedding-based retrieval using averaged patch-embeddings from PathSSL is also provided for comparison.

| Dataset | Model | MAP | NDCG | Top-1 | Top-5 | Top-10 |
|---------|-------|-----|------|-------|-------|--------|
| DS1 ($N = 4,876$) | PathSSL | 0.23 | 0.47 | 0.29 | 0.48 | 0.57 |
| | PathAlign-R | 0.26 | 0.50 | 0.27 | 0.52 | 0.63 |
| DS1 ($N = 3,264$) | PathSSL | 0.29 | 0.67 | 0.62 | 0.83 | 0.88 |
| | PathAlign-R | 0.40 | 0.72 | 0.60 | 0.84 | 0.89 |

Table C.7: **Text inputs used for embedding-based image-classification.** The set of texts associated with each class consists of all combinations between task-specific prefixes and class-specific suffixes. The label in the *Class* column is being predicted for each task, and an ensemble of TASK PREFIX : CLASS SUFFIX is used to represent each class for the embedding-based classification.

| Task | Task prefix | Class | Class prefix | |
|---|---|---|---|---|
| NSCLC Sub-typing (TCGA) | lung, lobe, lobectomy
lung, lobe, wedge resection
lung, lobe, mass excision
lung, lobe, resection
lung, resection | LUAD | adenocarcinoma
lung adenocarcinoma
lung adenocarcinoma mixed subtype | |
| | | LUSC | squamous cell carcinoma
lung squamous cell carcinoma | |
| RCC Sub-yping (TCGA) | kidney, nephrectomy
kidney, resection | KICH | chromophobe renal cell carcinoma
renal cell carcinoma, chromophobe type
renal cell carcinoma of the chromophobe type | |
| | | KIRC | kidney clear cell renal cell carcinoma
kidney clear cell renal carcinoma
kidney renal clear cell carcinoma
renal cell carcinoma, clear cell type
renal cell carcinoma, clear cell type (conventional) | |
| | | KIRP | kidney papillary renal cell carcinoma
kidney renal papillary cell carcinoma
papillary renal cell carcinoma
papillary renal cell carcinoma (chromophil)
renal cell carcinoma, papillary type
renal cell carcinoma of the papillary type | |
| BRCA Sub-yping (TCGA) | breast, lumpectomy
breast, mastectomy
breast, excision
breast, resection | IDC | infiltrating ductal carcinoma
invasive ductal carcinoma
breast invasive ductal carcinoma
invasive ductal carcinoma of the breast
invasive carcinoma of the breast, ductal pattern | |
| | | ILC | infiltrating lobular carcinoma
invasive lobular carcinoma
breast invasive lobular carcinoma
invasive lobular carcinoma of the breast
invasive carcinoma of the breast, lobular pattern | |
| Procedure type (DS1) | N/A | Biopsy | biopsy | |
| | | Resection | resection
lobectomy
gastrectomy
colectomy
pancreatectomy
appendectomy
oophorectomy
salpingo-oophorectomy
prostatectomy
lumpectomy
esophagectomy
orchiectomy | wedge resection
pneumonectomy
nephrectomy
splenectomy
cholecystectomy
thyroidectomy
salpingectomy
hysterectomy
mastectomy
hepatectomy
cervicectomy |

Table C.8: **Pathologist rating rubric.** Pathologist raters reviewed the top 5 retrieved texts, the generated text, and the original diagnostic text. Based on review of the associated WSI, texts were scored on a scale of 1 to 5, with an option to indicate issues with quality or the need for more information. The set of texts per WSI were presented in random order and the raters were blinded to whether texts were retrieved, generated, or original.

| Rating | Description and instructions |
|---|---|
| 1 | **Completely inaccurate**
– May describe something that can occur in the specimen/tissue type pictured, but fundamentally incorrect, or may be the wrong tissue type or concept altogether. |
| 2 | **Partially accurate (*i.e.* related but wrong)**
– The text might describe an entity that is related to the image, and occurring in that specimen type, but the image is definitively a different diagnostic entity.
– May accurately describe something that is seen on the image, but additional, essential info is missing or incorrect. |
| 3 | **Mostly accurate *with* clinically significant error/omission**
– The text is a good match/description for the image, but something minor is incorrect or missing that may have clinical or diagnostic implications. |
| 4 | **Mostly accurate *without* clinically significant error/omission**
– The text is a very good match/description for the image, but there may be a minor, clinically insignificant aspect that is incorrect or missing. For example, the diagnosis is accurate and acceptable, but doesn't capture all of the details. |
| 5 | **Highly accurate**
– The text is a great description of the image, with no obvious information missing or incorrect.
– Note that even a very short summary or a description of "no pathologic findings" can still belong in this rating. |
| Cannot Interpret | Please provide a very brief comment regarding the issue and/or what additional info you would need.
If you can interpret the image to some extent, but need IHC or other studies to be more confident, please still provide a score based on your best interpretation of the available image and provide details in the comments. |

Table C.9: **Categories for evaluation of generated text.** As used in Figure 3b.

| Category | Generated text rating | Original text rating |
|---|---|---|
| AI preferred | $\geq 4$ | $\leq 3$ |
| Both ok, AI preferred | 5 | 4 |
| Both ok, same rating | 4 | 4 |
|  | 5 | 5 |
| Both ok, original preferred | 4 | 5 |
| Original preferred | $\leq 3$ | $\geq 4$ |
| Both with errors or omissions | $\leq 3$ | $\leq 3$ |

.

Table C.10: **Slide prioritization.** A set of colon biopsies was randomly sampled ($N = 200$), roughly representing a theoretical case load. We then prompted the LLM to generate a *prioritization score* for each image, which in turn is used to sort the images. The sorted results for all 200 biopsies are shown (note that original diagnostic text was not used for prompting, it is only provided here for reference). WSIs with identical diagnostic texts are grouped with counts for that text in the *count* column. While not perfect, this sorted list highlights the potential for flexible, user-prompted case prioritization. The LLM is prompted with each image, the generated text, and the following prompt: `Question:  On a scale of 1 to 3, where 1 is benign or low-risk, 2 are pre-cancerous polyps and adenomas, 3 is cancerous or highly suspicious for cancer, can you rate the pathological findings for this image?  Answer:`

| Original findings | AI prioritization | Count |
| --- | --- | --- |
| tubular adenoma with high grade dysplasia and focal intramucosal carcinoma. | 3 | 1 |
| invasive moderately differentiated adenocarcinoma of the colon. | 3 | 1 |
| invasive poorly differentiated colonic adenocarcinoma. | 3 | 1 |
| fragment of ulcer debris. - no colonic mucosa identified. | 3 | 1 |
| adenomatous polyp. | 2 | 26 |
| tubular adenoma. | 2 | 5 |
| multiple fragments of flat and polypoid colonic mucosa with adenomatous epithelium, consistent with multiple colonic adenomas. | 2 | 1 |
| adenomatous polyp with focal high grade dysplasia and trauma related changes. | 2 | 1 |
| mild active colitis, no evidence of chronicity. | 2 | 1 |
| adenomatous polyps. | 2 | 1 |
| essentially unremarkable colonic mucosa. | 2 | 1 |
| tubular adenoma. - negative for high grade dysplasia or carcinoma. | 2 | 1 |
| tubular adenoma fully excised in the sections examined. | 2 | 1 |
| chronic colitis with moderate to severe activity. no dysplasia identified. | 2 | 1 |
| adenomatous polyp, low grade. | 2 | 1 |
| adenomatous polyp (tubular adenoma). electrocautery margin appears uninvolved. | 2 | 1 |
| fragments of adenomatous polyp. | 2 | 1 |
| active chronic colitis with crypt abscess. | 2 | 1 |
| colonic mucosa with no pathologic diagnosis. | 1 | 16 |
| colonic mucosa with no significant pathologic abnormality. | 1 | 14 |
| hyperplastic polyp. | 1 | 12 |
| unremarkable colonic mucosa. | 1 | 8 |
| essentially unremarkable colonic mucosa. | 1 | 7 |
| benign colonic mucosa with no significant pathologic abnormality. | 1 | 6 |
| colonic mucosa with no significant pathologic changes. | 1 | 5 |
| tubular adenoma. | 1 | 5 |
| colonic mucosa with no diagnostic alteration. | 1 | 4 |
| chronic active colitis. | 1 | 4 |
| colonic mucosa with no pathologic diagnosis; negative for dysplasia. | 1 | 4 |
| colonic mucosa with no significant microscopic abnormality. | 1 | 3 |
| polypoid fragment of benign colonic mucosa. | 1 | 2 |

| | | |
|---|---|---|
| no significant abnormalities. | 1 | 2 |
| benign colonic mucosa with no diagnostic abnormality. | 1 | 2 |
| benign colonic mucosa with no diagnostic alteration. no dysplasia identified. | 1 | 2 |
| colonic mucosa overlying lymphoid aggregates otherwise no significant microscopic abnormality. | 1 | 2 |
| fragments of benign colonic mucosa. | 1 | 2 |
| unremarakble colonic mucosa. | 1 | 2 |
| colonic mucosa with no diagnostic alteration; negative for dysplasia. | 1 | 2 |
| active colitis with non-necrotic granulomas and features of remote and persistent injury. | 1 | 2 |
| chronic inactive colitis. | 1 | 2 |
| sessile serrated polyp. | 1 | 2 |
| tubular adenomas (2). - negative for high grade dysplasia or carcinoma. | 1 | 1 |
| tubular adenoma with surface cautery artifact. | 1 | 1 |
| diminutive adenomatous polyp. | 1 | 1 |
| focal active colitis. | 1 | 1 |
| chronic inactive colitis. - no dysplasia or granulomas identified. | 1 | 1 |
| benign colonic mucosa with no significant microscopic abnormality. | 1 | 1 |
| colonic quiescent colitis with hyperplastic change; negative for dysplasia. | 1 | 1 |
| polypoid fragments of benign colonic mucosa. | 1 | 1 |
| benign colonic mucosa with rare clusters of neutrophils in the lamina propria. - no chronic architectural changes, granulomas or dysplasia identified. | 1 | 1 |
| unremarkable colonic mucosa with increased eosinophils; likely due to medication. | 1 | 1 |
| tubular adenoma. - negative for high grade dysplasia or carcinoma. | 1 | 1 |
| no diagnostic alteration. | | 1 |
| quiescent colitis with focal hyperplasia. no dysplasia identified. | 1 | 1 |
| colonic mucosa and fibroadipose submucosa with no pathologic diagnosis. | 1 | 1 |
| features consistent with submucosal lipoma. | 1 | 1 |
| tubular adenoma. - no high grade dysplasia or carinoma identified. | 1 | 1 |
| consistent with hyperplastic polyps (2). | 1 | 1 |
| benign colonic mucosa with no significant pathology. | 1 | 1 |
| benign colonic mucosa with no pathologic diagnosis. | 1 | 1 |
| colonic mucosa with mild crypt architectural distortion; no dysplasia identified. | 1 | 1 |
| inactive chronic crypt destructive colitis without granulomas; no dysplasia identified. | 1 | 1 |
| hyperplastic polyps (2). fragments of unremarkable colonic mucosa (3). | 1 | 1 |
| benign colonic mucosa with hyperplastic change. | 1 | 1 |
| benign polypoid fragment of colonic mucosa with no microscopic abnormality. | 1 | 1 |
| adenomatoid polyp. | 1 | 1 |
| colonic mucosa with no pathologic changes. | 1 | 1 |
| tubular adenoma(s). | 1 | 1 |
| colonic mucosa with glandular architectural changes consistent with chronic inactive colitis. - negative for dysplasia. | 1 | 1 |
| hyperplastic polyp. colonic mucosa with lymphoid aggregate formation. | 1 | 1 |
| fragments of unremarkable colonic mucosa. | 1 | 1 |

| | | |
|---|---|---|
| benign colonic mucosa with focal hyperplastic changes. | 1 | 1 |
| tubular adenoma. - benign colonic mucosa. | 1 | 1 |
| colonic mucosa with benign lymphoid aggregates and no pathologic diagnosis. | 1 | 1 |
| benign colonic mucosa with prominent lymphoid aggregate. | 1 | 1 |
| fragments of colonic mucosa with hyperplastic change. | 1 | 1 |
| polypoid fragment of colonic mucosa with lamina propria edema, fibrosis and mild chronic inflammation. | 1 | 1 |
| portions of colonic mucosa with no significant microscopic abnormalities. | 1 | 1 |
| focal acute inflammation. | 1 | 1 |
| colonic mucosa with mild architectural disorder. - negative for dysplasia. | 1 | 1 |
| fragments of colonic mucosa with no significant pathologic changes. | 1 | 1 |
| portions of colonic mucosa with pigmented macrophages in the lamina propria. | 1 | 1 |
| benign colonic mucosa with no pathologic abnormality. | 1 | 1 |
| unremarkable colonic/rectal mucosa. | 1 | 1 |

Table C.11: **Overview of TCGA with cases split by tissue source site (TSS) to create held out TSS in validation and test splits.** Within each study, TSS codes were sorted by number of cases from each site (noting that codes for the same TSS are not the same codes across studies). Entire TSSs were assigned to the train split until the train split contained at most 55% of cases. The remaining TSSs were then assigned to validation and test splits in an alternating fashion.

| Study | Split | TSS code | #Cases | #Slides |
|---|---|---|---|---|
| ACC | train | OR [1] | 82 | 201 |
| | validation | PA, P6 | 4 | 2 |
| | test | PK, OU | 6 | 24 |
| BLCA | train | XF, ZF, DK, FD, BT | 206 | 188 |
| | validation | FJ, SY, E5, 5N, K4, 2F, LT, GU, BL, H4, E7, CU, LC, R3, UY | 98 | 117 |
| | test | G2, S5, YF, 4Z, CF, YC, HQ, FT, PQ, GV, GD, KQ, C4, MV, GC | 108 | 153 |
| BRCA | train | A2, E2, AR, A8, D8, BH | 574 | 605 |
| | validation | PE, XX, AQ, HN, UU, MS, PL, A1, EW, GM, 5T, GI, AN, W8, AC, OK, B6 | 250 | 238 |
| | test | OL, 4H, LL, LQ, WT, S3, 3C, UL, Z7, V7, JL, E9, C8, A7, LD, 5L, AO | 273 | 284 |
| CESC | train | VS, EK, C5 | 146 | 116 |
| | validation | LP, R2, RA, XS, 4J, BI, HM, EX, PN, ZX, IR, 2W, DR, DS, WL, JX, ZJ, HG, GH | 77 | 74 |
| | test | FU, DG, Q1, UC, MU, MY, EA, MA, JW | 84 | 89 |
| CHOL | train | W5 | 21 | 18 |
| | validation | ZU, 3X, 4G, ZD | 12 | 9 |
| | test | YR, ZH, W6, WD | 12 | 12 |
| COAD | train | AA, A6 | 225 | 781 |
| | validation | AU, QG, RU, SS, QL, DM, AY, D5, 4T, F4, WS, 3L, AD | 107 | 266 |
| | test | CK, CM, AM, 4N, G4, CA, AZ, 5M, NH, T9 | 125 | 363 |
| DLBC | train | FF, FA | 12 | 16 |
| | validation | RQ, G8 | 6 | 6 |
| | test | GS, VB, FM, GR | 7 | 10 |
| ESCA | train | LN, L5 | 86 | 59 |
| | validation | V5, M9, ZR, R6, XP, IC, L7, Q9, X8, RE, VR, KH | 49 | 38 |
| | test | JY, S8, IG, 2H, Z6 | 50 | 50 |
| GBM | train | 06, 12, 02 | 307 | 564 |
| | validation | 15, 4W, 87, 26, 76, 28, 41, 19 | 133 | 100 |
| | test | 14, 32, 81, 27, 16, RR, OX, 74, 08 | 155 | 197 |
| HNSC | train | CQ, CV, CN | 249 | 234 |
| | validation | RS, 4P, BB, IQ, C9, DQ, P3, UF, MZ, HL, H7, KU | 100 | 97 |
| | test | T3, HD, D6, T2, BA, MT, QK, TN, CX, WA, UP, F7 | 125 | 141 |

1. One predominant TSS for this study and split.

| | | | | |
|---|---|---|---|---|
| KICH | train | KL, UW | 49 | 37 |
| | validation | KN, NP | 26 | 26 |
| | test | KO, KM | 38 | 23 |
| KIRC | train | BP, B0 | 249 | 251 |
| | validation | A3, T7, DV, B2, MW, 6D, GK, G6, 3Z, EU, CW, MM, B8 | 141 | 127 |
| | test | AS, AK, CZ, CJ, B4 | 147 | 147 |
| KIRP | train | A4, 5P, B9, UZ, SX, BQ, 2Z | 152 | 150 |
| | validation | F9, HE, IZ, B1, UN, P4, IA, WN, DW, AT, O9, PJ, 4A | 65 | 61 |
| | test | AL, Y8, MH, Q2, V9, G7, EV, GL, 2K, B3, DZ, KV, J7 | 73 | 87 |
| LGG | train | HT, S9, FG, DU | 279 | 487 |
| | validation | HW, FN, KT, WY, E1, WH, VM, IK, TM, QH, VW | 107 | 147 |
| | test | CS, DH, F6, DB, VV, P5, RY, TQ, EZ, R8, W9 | 129 | 177 |
| LIHC | train | G3, DD | 184 | 187 |
| | validation | O8, BC, RG, YA, NI, RC, 5R, K7, ED, WJ, T1, 3K, 4R, XR, 2V, PD, BW, WX, MR, QA, ZS, ES, EP | 90 | 88 |
| | test | ZP, 5C, KR, LG, 2Y, UB, HP, FV, WQ, CC, BD, GJ, MI | 103 | 97 |
| LUAD | train | 05, 50, 44, 78, 86, 55 | 273 | 248 |
| | validation | 75, 91, 99, 64, MN, 4B, 95, 97, S2, L9, 67, 35, 71 | 115 | 100 |
| | test | 80, 53, MP, 93, O1, 73, 69, 49, NJ, L4, 83, 62, J2, 38 | 134 | 183 |
| LUSC | train | 60, 22, 66, 85, 63, 56, 77 | 260 | 260 |
| | validation | 39, XC, 6A, O2, 68, 52, MF, 98, 34, 70, 90, 18, 51, 58 | 117 | 97 |
| | test | 46, LA, 33, 43, L3, 37, NC, NK, 79, J1, 96, 21, 94, 92 | 127 | 156 |
| MESO | train | TS, 3H, MQ, LK | 44 | 46 |
| | validation | NQ, SC, UT, ZN | 20 | 26 |
| | test | YS, 3U, SH, XT, UD | 23 | 23 |
| OV | train | 13, 61, 24 | 274 | 2 |
| | validation | 42, 57, 25, VG, 10, 36, 20, 23, 5X, 3P | 152 | 100 |
| | test | OY, 30, 09, WR, 29, 59, 31, 04 | 161 | 4 |
| PAAD | train | IB, 2J, HZ | 86 | 88 |
| | validation | YH, M8, XN, 3A, H6, US, YB, H8, L1, RL, XD, LB, HV, YY | 49 | 69 |
| | test | FB, 3E, RB, FZ, 2L, OE, PZ, S4, Z5, F2, Q3 | 50 | 48 |
| PCPG | train | QR, WB | 82 | 83 |
| | validation | SQ, RM, P7, RX, SP, XG, P8, W2, S7, PR | 48 | 61 |
| | test | QT, TT, RW, SA, SR, RT | 49 | 51 |

| | | | | |
|------|------------|-----------------------------------------------------------|-----|-----|
| PRAD | train | HC, KK, EJ, G9 | 251 | 217 |
| | validation | X4, YJ, VP, TK, SU, VN, Y6, 2A, V1, M7, HI, FC, XA, ZG | 115 | 100 |
| | test | H9, J9, WW, J4, CH, TP, 4L, XJ, QU, XQ, YL, XK, MG, KC | 134 | 126 |
| READ | train | AG | 80 | 72 |
| | validation | DY, G5, DT, F5, BM, AF, CL | 42 | 42 |
| | test | EF, EI, CI, AH, DC | 45 | 40 |
| SARC | train | 3B, DX | 137 | 360 |
| | validation | HB, SG, RN, KF, UE, Z4, IE, QQ, KD, PT, IW, X2, X9, WK, VT, SI | 59 | 117 |
| | test | K1, LI, PC, QC, MO, N1, X6, WP, 3R, FX, HS, IS, IF, MB, JV, MJ | 64 | 118 |
| SKCM | train | EB, EE | 136 | 136 |
| | validation | FR, W3, QB, BF, IH, HR, WE, YD, RP, LH, D9, 3N, FS, D3, GF, YG, Z2 | 82 | 87 |
| | test | ER, FW, XV, GN, DA | 82 | 83 |
| STAD | train | BR, VQ | 205 | 180 |
| | validation | MX, ZQ, HF, ZA, RD, SW, EQ, HJ, CD, IP, R5, KB, CG | 117 | 91 |
| | test | F1, D7, B7, HU, FP, IN, 3M | 121 | 122 |
| TGCT | train | 2G | 64 | 113 |
| | validation | 2X, SB, XY, SO, 4K, VF, YU, X3, W4 | 34 | 52 |
| | test | S6, SN, XE, ZM, WZ | 36 | 38 |
| THCA | train | EL, EM, DJ | 254 | 260 |
| | validation | E8, DO, IM, 4C, KS, L6, BJ, DE, FK | 117 | 123 |
| | test | FE, E3, CE, MK, FY, J8, H2, GE, QD, ET | 136 | 136 |
| THYM | train | X7, ZB, XU | 66 | 65 |
| | validation | 4V, 3Q, ZC, 3S, 3G, 5V, ZT, 3T | 27 | 30 |
| | test | XM, XH, 5G, ZL, 5U, 4X, 5K, YT | 31 | 85 |
| UCEC | train | A5, D1, AX, AP, B5 | 294 | 290 |
| | validation | KP, FI, AW, KJ, PG, 2E, EY, BS, DI, SJ, JU, EC, 5B | 113 | 152 |
| | test | QS, EO, H5, QF, 5S, 4E, BK, K6, SL, BG, DF, AJ, E6 | 141 | 147 |
| UCS | train | N5, N8, NA | 22 | 50 |
| | validation | NG, N9, QN, NF | 13 | 31 |
| | test | N6, QM, N7, ND | 15 | 50 |
| UVM | train | V4 | 33 | 33 |
| | validation | V3, WC | 16 | 16 |
| | test | RZ, YZ, VD | 31 | 23 |

