# OpenReview forum: "PathAlign:  A vision--language model for whole slide images in histopathology"
_MICCAI.org/2024/Workshop/COMPAYL — COMPAYL 2024_

### Official Review · Reviewer_gruK · 2024-07-02
**Interesting VLM for histopathology, good results, lack of novelty**

**Custom Rating:** 4
**Confidence:** 4

**Review:**

Summary:
The paper tackles text-to-image alignment between patches and medical reports (text) on a WSI level.

Strengths:
- Experiments are extensive
- The paper is well-written and describes the model well

Weakness:
- motivation and novelty are a bit unclear.
- The underlying VLM model is not novel

Conclusion:
Despite the unclear contribution, the paper describes a good VLM solution for histopathology and should be published.

---

### Official Review · Reviewer_2Xxr · 2024-07-03
**A vision-language models for WSI**

**Custom Rating:** 3
**Confidence:** 2

**Review:**

The authors present a visual language model based on BLIP-2 trained on 350,000+ WSI with paired text descriptions. Model output has been verified by pathologist and ranked as accurate for 78% of the slides.

Below are my comments:
1. The introduction mentions some limitations of the literature but without citing backing evidence.
2. The literature review section is very limited in size and context.
3. The contributions of the work are not concretely outlined. Yes, this is visual-language model, and yes, this is a hot research topic at the moment, but rigorous contributions and points of novelty are still necessary in research publications, in my opinion.
4. Since this work involves a human-based evaluation by comparing original and AI-generated description, I believe it's important to show how a rating is computed, i.e., by showing the user interface through which the pathologist carries out the reviews and how the options (human vs. AI) are presented. Again, this is important to mitigate bias.
5. Why only 120 WSIs are used for pathologist evaluation? Isn't this number too low? And how that subset is taken without introducing any further bias?
6. What is the technical contribution of the work? If it is based on BLIP-2, it is unclear to me how the work is not an application of an existing on model.
7. As far as I can see, the work has not been benchmarked (or compared) against similar studies. This is quite pivotal in developing models that we can trust to use in the wild.

---

### Official Review · Reviewer_bmGi · 2024-07-10
**Timely and complete work**

**Custom Rating:** 5
**Confidence:** 4

**Review:**

This paper is on the very timely topic of vision-language models  in histopathology, merging both pathology reporting data and images. Authors rightfully highlight the important real-life setting of  multiple images  /case, with only a subset of these slides containing the diagnosis reported in the pathology report. The presented model-generated texts are further evaluated by pathologists and found to be promising. A couple of minor points to complement this nice paper:
Were all WSI’s in the DS1 cancer diagnoses / suspicion for cancer diagnosis and does this information also come from the text (i.e. suspicion for …, is information from clinicians also transferred or found within the pathology reports?). What is the disctribution of tissue types?  Could the authors comment on why BLIP-2 framework was used rather than the others mentioned? Perhaps a word in the main text would be useful. What was the inter-observer variability between pathologists? Could the authors specify what “subtyping” refers to for their tasks? Will the task be simplified with the continued adoption of structure / Synoptic reporting by pathologists, or will this data structuring task become redundant?

---

### Decision · Program_Chairs · 2024-07-16

Accept